# Analysis of Biometric-Based Cryptographic Key Exchange Protocols—BAKE and BRAKE

**Maksymilian Gorski and Wojciech Wodo \***

Faculty of Information and Communication Technology, Wroclaw University of Science and Technology, Wybrzeze Wyspianskiego 27, 50-370 Wroclaw, Poland; 259396@student.pwr.edu.pl
\* Correspondence: wojciech.wodo@pwr.edu.pl

**Abstract:** Biometric authentication methods offer high-quality mechanisms to confirm the identity of individuals in security systems commonly used in the modern world, such as physical access control, online banking, or mobile device unlocking. They also find their application in cryptographic solutions, which allow the biometrically authenticated exchange of cryptographic keys between users and services on the internet, despite the fuzziness of biometric data. Such solutions are BAKE (*biometrics-authenticated key exchange*) and BRAKE (*biometric-resilient authenticated key exchange*) protocols, upon which our work is based. However, the direct application of fuzzy biometrics in cryptography, which relies heavily on the accuracy of single-bit secret values, is not trivial. Therefore, this paper is devoted to analyzing the security of this idea and the feasibility of implementing biometric AKE (*authenticated key exchange*) protocols, with an emphasis on the BRAKE protocol. As the results of our analysis, we discuss BRAKE's limitations and vulnerabilities, which need to be appropriately addressed to implement the protocol in modern systems.

**Keywords:** cryptography; biometrics; key exchange; protocol; fuzzy vault; OPRF; AKE; BAKE; BRAKE





## 1. Introduction

Over the last few years, there has been a growing interest in the use of biometric authentication methods in the creation of cryptographic solutions. This is due to the fact that authentication based on biometric features allows one to confirm the identity of a given person, with high confidence. This property is especially desirable in real-world applications that present a need for high-trust authentication to confirm the identity of its users, such as physical access control and digital electronic banking systems.

A frequently reported problem with modern digital authentication methods is the lack of confirmation of a specific person's identity. This is because authentication is usually based on the value of a secret key held by the user and stored in the persistent memory of that user's device. This approach does not directly confirm the identity of the physical user, only the fact that an individual is in possession of the secret key, which is associated with that person. In addition, using knowledge of the secret key value as the main authentication factor exposes the lack of confirmation of the user's intention to participate in a cryptographic protocol. Thus, in the scenario when the authentication key value is stolen due to an attack against the user's device, the adversary can correctly complete the authentication process on behalf of the victim, which is not possible with the use of biometric-based authentication methods.

The authors of the BAKE (*biometrics-authenticated key exchange*) [1] and BRAKE (*biometric resilient authenticated key exchange*) [2] protocols address this problem by presenting schemes that allow for the generation of secret authentication keys directly from the biometric modalities of users. These keys are characterized by the lack of the need to store their values in the persistent memory of devices, as they ought to be used only during the execution of the protocols and discarded afterward. This allows us to take advantage of

the benefits of authenticating a person using their biometric features and provides strong evidence of their willingness to participate in the protocol. This happens due to the need to present the biometric modality to the measuring device (e.g., scanner), which is most often part of the user's terminal (e.g., mobile device or smartphone).

The development of such solutions is not a trivial task due to the requirement of successfully including fuzzy biometric data into the cryptographic solutions, which in most cases require single-bit accuracy of provided and processed secret values. Fortunately, in recent years, major steps were taken in the context of merging the fields of biometrics and cryptography to ensure strong authentication methods while preserving users' sensitive data and using them as essential parts of cryptographic mechanisms, as shown in [3–5].

However, despite many years of research on combining the use of biometrics-based methods with cryptographic systems, the protocols presented in [1,2] were of particular interest to us. These are the only mechanisms that we were able to reach, which focus not only on creating modular functionalities, combining cryptography with biometrics, as is the case with fuzzy vault [4], but also on presenting full solutions with the potential to replace real, currently used protocols. Due to the above-mentioned novelty and the way various cryptographic functionalities are implemented into the discussed protocols, we decided to conduct a security analysis of the proposed solutions in this work, focusing on the BRAKE protocol, as its creators undertook an extensive analysis and detection of limitations of the BAKE scheme in [2].

*Contribution*

This work highlights the results of the analysis proposed in the work [2] on BRAKE protocol, the successor of the BAKE protocol described in [1]. The analysis was devoted to finding security vulnerabilities (both in design and implementation) and revealed some flaws in the protocol, particularly the following:

- The protocol's lack of resistance to the compromise of the evaluator's secret key, used as part of the OPRF (*oblivious pseudo-random function*) primitive in the registration and verification processes of users. This may lead to the execution of an offline attack on a specific user's biometric template and, as a consequence, compromise the asymmetric cryptography keys used in the authentication process.
- A risk related to the storage of secret values, such as the coefficients of the secret polynomial $f(x)$ and the secret keys $csk_t$ and $csk_{t'}$ in the persistent memory of the user's terminal. This may lead to the successful passing of the user authentication process, despite the provision of a non-mated biometric characteristic.
- An unauthorized adversary may be able to interrupt already-established communication sessions between users and the server. This may be conducted by requesting the server to start a new verification process on behalf of specific users, which may result in the generation of a new session key and revocation of the previously used keys, preventing legitimate users from the continuation of the current session.
- An unauthorized adversary may be able to perform a *denial-of-service* attack on a server instance by sending to the server a significant number of requests to register new user identities, using falsified biometric data. This could populate the server's identity database with entries for non-existent users.

In order to accurately present the reasoning behind the analysis, it is necessary to first understand the idea behind the emergence and potential use of the BAKE [1] and BRAKE [2] protocols. Thus, we also provide brief descriptions of the protocols in the order they were proposed and published. It is also important to highlight that the latter one—the BRAKE protocol—has been inspired by the work of BAKE protocol authors, as it takes into consideration the potential limitations that threaten the declared security of BAKE.

## 2. BAKE Protocol Overview

According to its creators, the motivation behind the design of the protocol was to create a mechanism that would allow the use of end-to-end encryption in peer-to-peer

communication between client devices. An important aspect of the resulting mechanism is the introduction of authentication functionality that is an integral part of the protocol, using the user's biometric features for this purpose. Secret values used in the protocol, such as secret asymmetric cryptography keys, are derived directly from the result of measuring the user's biometric modality. The correctness of their calculation depends on the sufficient proximity of the provided biometric modality to the one provided in the key generation phase of the protocol.

As part of the BAKE protocol, the AFEM (*asymmetric fuzzy encapsulation mechanism)* has been proposed, which is a scheme that describes the order of communication between the parties during the execution of the protocol. The mechanism is divided into three phases:

- *Initialization phase:* responsible for the configuration of the protocol and establishing the values of public parameters used in communication between the parties.
- *Key generation phase:* where asymmetric cryptography key pairs are generated and then distributed between the parties.
- *Authenticated key exchange phase:* where the value of the symmetric session key is established between the parties.

A breakdown of the steps required to be performed during each phase is depicted in Figure 1, which is based on the scheme presented by the authors. According to the assumptions of the use case of the protocol, the initialization and key generation phases ought to be executed only once for every pair of users. In order to maintain the legibility of the presented scheme, we present a description of the mathematical notation of the used parameters:

- $\lambda$—security bits of the executed protocol.
- $\tau$—desired level of closeness between the biometric templates provided during the key exchange phase and the reference template provided in the key generation phase.
- $sk, sk'$—reference and query asymmetric secret keys, respectively.
- $pk$—asymmetric public key derived directly from a given secret key.
- $s$—randomly generated secret message.
- $c$—encapsulated form of the secret message, $s$.
- $s'$—secret message obtained through the decapsulation of the ciphertext, $c$.
- $k$—established session key.
- $H(\cdot) : \{0,1\}^* \to \mathbb{Z}_q$—hash function used as the key derivation function.

The AFEM consists of four PPT (*probabilistic polynomial time*) algorithms: *AFEM.Setup*, *AFEM.PubGen*, *AFEM.Enc*, and *AFEM.Dec*. These PPT algorithms are, respectively, responsible for the following: the generation and distribution of public parameters to both parties; the generation of the asymmetric public key from the provided secret key; and the encapsulation and decapsulation of the message distributed between the communication participants.

To summarize the process of the protocol execution: after setting up the public parameters, $pp$, directly between users or with the help of a TTP (*trusted third party*) , both users generate their reference secret keys, $sk$, based on the biometric features provided. After the secrets are generated, each user distributes their public key generated using the *AFEM.PubGen* algorithm—this step ends the key generation phase. The authenticated key exchange phase starts with both participants randomly generating an ephemeral message, $s$, value, which is then encapsulated during the execution of *AFEM.Enc*. Next, both parties exchange the ciphertexts, $c$, of generated messages and again provide their biometric features to generate query secret keys, $sk'$. Each user is able to successfully decapsulate the message, $s'$, only if the measure of distance between the reference and query biometric characteristics fits in the $\tau$ threshold. Finally, the established session key, $k$, is locally calculated by both parties as the output of the hash function.

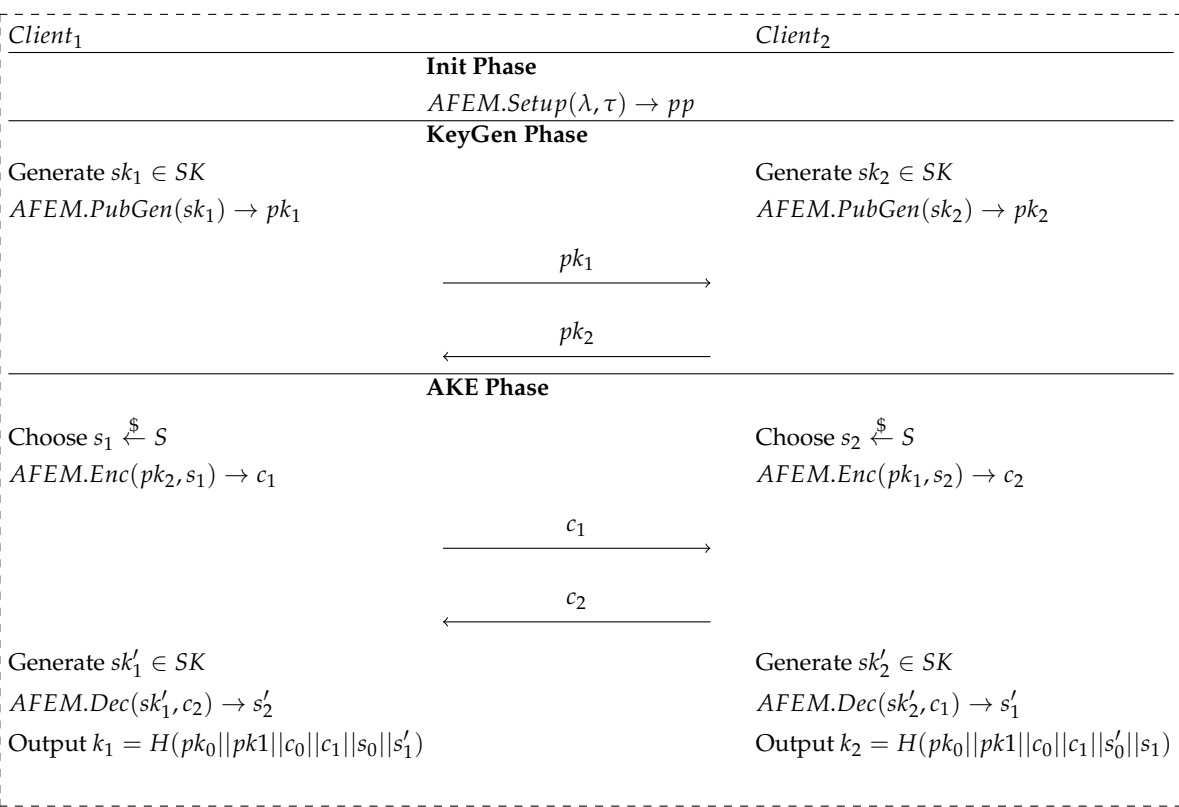

**Figure 1.** Scheme of the AFEM mechanism based on [1].

The presented mechanism is based on the Learning with Errors problem and uses two implementation methods, interchangeably used depending on the type of biometric modality from which the data are extracted: biometric vectors and biometric vectors set. Their comprehensive description is illustrated in [1]; thus, we refer the reader to the source work for more details. However, it is significant to highlight that each of the proposed implementations includes a definition of the distance function, $dis(\cdot, \cdot)$, used as a metric of closeness between the secret keys generated by one party. While working on the applications of the biometric characteristics in the development of cryptographic solutions, it is crucial to define metrics that allow one to decide whether the biometric data were extracted from a specific user, without compromising the features. The correctness of the BAKE framework is given by the following assumption:

$$Dec(sk', Enc(PubGen(sk, s))) = s \iff dis(sk, sk') < \tau \tag{1}$$

By the correctness given in Equation (1), it is easy to conclude that the distance threshold, $\tau$, determines the upper bound for the acceptable differences between the user's secret keys and, in consequence, the differences between fuzzy measurements of biometric features. A deep security and application analysis of the solutions suggested in [1] was undertaken in [2]; thus, we will not elaborate on it further in this work.

## 3. BRAKE Protocol Overview

In contrast to its predecessor, the authors of the BRAKE [2] protocol propose a communication model that is much more widespread in the context of modern computer networks than the one proposed in [1]. This is a client–server infrastructure in which a significant number of users communicate with a centralized server unit that stores the data of all client identities. An additional variation in the proposed infrastructure is the evaluator instance. It acts as a trusted third party, independent of the server instance, evaluating the values provided by clients as part of the implementation of the cryptographic primitive used in the

protocol—an *oblivious pseudo-random function*. It is worth mentioning that the creators of the protocol conducted an in-depth analysis of the BAKE protocol, identifying its limitations and potential security threats. Based on their findings, they proposed a scheme to mitigate the identified problems, while maintaining the security of the processed biometric data in accordance with the ISO/IEC 24745 standard.

As in the case of the BAKE protocol, the BRAKE protocol can be divided into two main phases, differing in the purpose and number of executions:

- *Enrolment phase*—responsible for creating and uploading a specific user's biometric identity to the server instance. During that process, the client provides the server with a public key derived from the reference biometric template, which can be used to determine the success of the user's authentication during the *verification phase*.
- *Verification phase*—this is performed each time the client attempts to establish a symmetric key that is used to encrypt the communication within the session. The client is only able to correctly establish the secret key with the server if the biometric authentication process is successful.

### 3.1. Interpretation of Biometric Features

When analyzing the BRAKE protocol, attention should be paid to the form in which the biometric template provided by the client is represented. The *fuzzy vault* primitive that was chosen by the protocol authors requires that the biometric features be represented in the form of a vector consisting of finite field elements with the order of the selected prime number, $t \in \mathbb{F}_q^n$. The authors presented schemes that meet the requirements for biometric data extracted from fingerprints, irises, and face scans. A detailed description of the extractors used is presented in [2]. An unquestionable advantage of BRAKE, compared to BAKE, is the fact that as long as it is possible to bring the biometric data to the indicated form, the protocol can be used regardless of the selected biometric modality, which translates into its high flexibility in the context of implementation.

### 3.2. Fuzzy Vault Primitive

To ensure secure storage and determine the sufficient proximity of the provided biometric features, the authors propose implementing a *fuzzy vault* [4] primitive into their scheme. It allows one to obfuscate the sensitive biometric features that are stored in the template, $t$, into the form of a finite field polynomial, $V \in \mathbb{F}_q[x]$. Along with the template, $t$, a small degree secret polynomial, $f \in \mathbb{F}_q[x]$, is locked in a vault. The required minimal closeness, $\tau$, of the reference and query templates, directly depends on the $f$ degree and is equal to $\tau = deg(f) + 1$.

The primitive's naming comes from the fact that secret values are locked in a vault in the form of a high-degree polynomial, $V$. It is possible to perform locking and unlocking operations on the vault, which correspond to the process of hiding and retrieving the secret values, respectively. Unlocking is possible only by providing sufficiently similar fuzzy secret values similar to those during the locking process. The way the vault polynomial is constructed allows it to be publicly distributed due to the difficulty of the unlocking operation without the appropriate secrets, as the biometric characteristics are used in this case. The functionalities are defined as follows: $lock(t) \to (f, V)$, $unlock(V, t') \to f'$.

The formal definition of the *fuzzy vault* primitive is given by the following:

$$V(x) = f(x) + \prod_{\forall a \in t} (x - a) \tag{2}$$

where polynomial $V(x) \in \mathbb{F}_q[x]$ is considered as the *fuzzy vault*, $f \in \mathbb{F}_q[x]$ is a secret polynomial, and $\prod_{\forall a \in t}(x - a)$ is a polynomial obtained by the multiplication of polynomials constructed using all values of the biometric vector, $t = [a_1, a_2, \ldots a_n] \in \mathbb{F}_q^n$. It is also important to highlight that without the addition of $f$, it would be possible to recover the biometric characteristics as the roots of the multiplied polynomial, which would compromise private user data.

The metric for assessing the similarity of biometric characteristics during the execution of the primitive is the intersection of the reference and query templates, $|t \cap t'|$. The $f'(x) = f(x)$ can be recovered properly only if both vectors, $t$ and $t'$, share a sufficient amount of values, noted as the biometric verification threshold, $\tau$. Thus, the correctness of the primitive is presented as follows:

$$f'(x) = unlock(V(x) = lock(t), t') = f(x) \iff |t \cap t'| \geqslant \tau \tag{3}$$

### 3.3. Oblivious Pseudo-Random Function Primitive

The *oblivious pseudo-random function* primitive and its variations were widely analyzed in [6], so in this work, we limit the primitive's description to the most important definitional concepts, which are as follows:

1. $F_k : \{0,1\}^\lambda \times \{0,1\}^m \to \{0,1\}^n$ is an *oblivious pseudo-random function* and $k \in \{0,1\}^\lambda$ is a secret key of the evaluator party.
2. $F_k(x)$ is efficiently computable from input $x$ provided by the client and the key, $k$, provided by the evaluator.
3. It is not possible to efficiently determine whether the primitive yielded the value, $F_k(x)$, for the given $x$ and $k$, or whether a random bit-string of length $n$ was returned.
4. The evaluator has no way of knowing the value of $x$ and the client has no way of knowing the value of the $k$ key, based on the value yielded by the primitive.

A generic scheme of the OPRF protocol is depicted in Figure 2. However, it is noticeable that the calculation of the value, $F_k(x)$, cannot take place between the client and the evaluator in a presented form, because no party can handle the actual computation. Therefore, the calculations need to be performed by the evaluator's instance. That is why the input value, $x$, provided by the client is obfuscated into the form of $[r]H(x)$, where $H : \{0,1\}^* \to \mathbb{G}_q$ is a cryptographic hash function, $\mathbb{G}_q$ denotes a cyclic group with the order of the prime number, $q$, and $[r]\cdot$ denotes the reversible obfuscation operation. In this situation, the evaluator has no way of learning the value of the input, $x$.

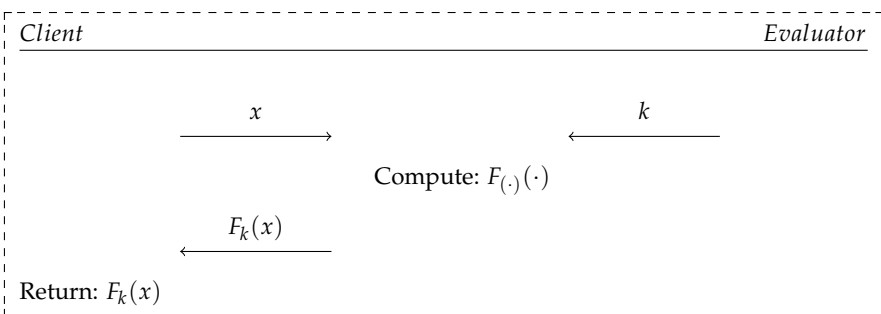

**Figure 2.** Generic scheme of the OPRF primitive execution.

According to the authors of [2], the OPRF primitive was introduced into the BRAKE protocol in order to hold control over the authentication requests sent from a specific client by enforcing interactions with every attempt. This feature allows exposing brute force attacks against the user's secret data by detecting a large quantity of requests sent on behalf of a specific client, in a short period of time. The need for interactions should also prevent the execution of offline attacks against the secret values of the victim (e.g., biometric template, $t$, secret polynomial, $f$, coefficients).

The use of the OPRF primitive in the BRAKE protocol is possible by introducing three algorithms:

- $blind(x) \to (B, r)$—an algorithm that obfuscates the value of $H(f)$ used as the $x$ argument provided by the client using a randomly generated value, $r \in \mathbb{Z}_q$, to the obfuscated form of $B \leftarrow [r]H(f)$.

- $eval(B,k) \rightarrow S$—an algorithm that evaluates the value of $S \leftarrow [k]B$ within the OPRF primitive on the evaluator's instance, where $k$ is the secret key known only by the evaluator;
- $unblind(S,r) \rightarrow U$—an algorithm that deobfuscates the value of $S$—obtained as a result of the evaluation—into the form of $U \leftarrow [r']S$, where $r' \in \mathbb{Z}_q$ is the inverse element of $r'$.

The method of using the above mechanisms is shown in Figure 3. The value received by the client as a result of running the protocol is $[k]H(f)$, in accordance with the equality, as follows:

$$U = [r']S = [r']([k]B) = [r']\Big([k]\big([r]H(f)\big)\Big) = [k]H(f) \tag{4}$$

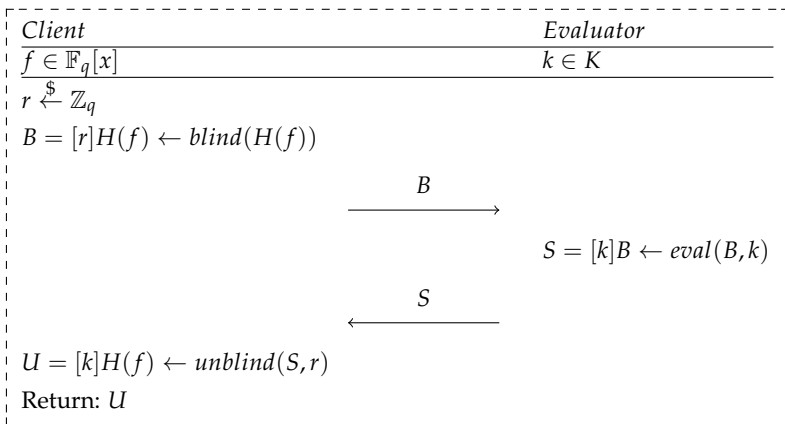

**Figure 3.** Scheme of the OPRF scheme implemented in the BRAKE protocol.

By obfuscating and deobfuscating the value of $H(f)$, the client's input privacy assumption of the OPRF primitive is fulfilled. Despite the evaluation performed by the evaluator's instance, it does not obtain any information about the coefficients of the secret polynomial, $f$, as well as the value of the hash function, $H(f)$. Simultaneously, the client is not able to efficiently determine the value of the key, $k$, used in the evaluation process based on the yielded result.

*3.4. BRAKE Enrolment Phase*

The presented scheme is based on a one-time execution of the *enrolment phase*, during which, the client provides a biometric sample, also denoted as a reference template, $t$. Using the reference template, an asymmetric cryptography key pair is created. These keys are later used to establish the session keys between the client and the server in the *verification phase*.

In Figure 4, we can see the use of each primitive described in Sections 3.2 and 3.3. First, the biometric vector, $t$, obtained through the measurement and processing of the modality, is used to construct a public polynomial, $V$, using the randomly generated secret polynomial $f$ of degree $deg(f) = \tau - 1$. This process is followed by the creation of an asymmetric key pair $(csk_t, cpk_t)$, of which the public key, $cpk_t$, will be transferred to the server's database. The OPRF protocol is executed between the client and evaluator, and the result of the evaluation process is treated by the client as the value of the private key, $csk_t$. Next, the algorithm for generating the public key, $pkGen()$, is carried out, based on the value of the private key, $csk_t$, known only to the client. The client also has to choose an identification value, $id$, which is used in the verification process. A tuple of public parameters $(V, cpk_t, id)$ is sent to the server via an authenticated communication channel, where it is saved into the server's database.

**Figure 4.** Scheme of the *enrolment phase* introduced in BRAKE, based on [2].

The *enrolment phase*, in order to be correctly executed, introduces an additional $pkGen()$ algorithm, which is used for creating a public key, $cpk_t$, which corresponds to the asymmetric secret key, $csk_t$, obtained as an outcome of the OPRF result's deobfuscation. The key pair created this way is used in the key encapsulation mechanism implemented by the authors and, according to their assurances, a wide range of solutions can be used for this purpose, such as elliptic curve Diffie–Hellman, RSA, or CRYSTALS-Kyber.

*3.5. BRAKE Verification Phase*

The *verification phase*, as depicted in Figure 5, refers to any attempt to perform the authenticated exchange of a session key, $\rho$, with the server. During the phase, the client provides the measurement of the selected biometric modality, which is represented as a query template, $t'$. For successful authentication, the client attempts to reproduce the asymmetric private key value, $csk_{t'}$, such that $csk_{t'} = csk_t$. The key exchange process is possible only if the threshold of closeness, $\tau$, of the query template, $t'$, sufficiently similar to the reference template, $t$, is exceeded, which makes the $|t \cup t'| \geqslant \tau$ requirement fulfilled. In addition to executing the primitives presented in Sections 3.2 and 3.3, this phase also involves distributing the session key to the client by the server instance. The *verification phase* also introduces an ephemeral key generation algorithm $KeyGen()$, which yields asymmetric keys that are used for communication channel authentication. The public ephemeral key, $cpk_e$, is also shared with the server as it is used as one of the arguments for the key derivation function. The session key, $\rho$, is generated using a key derivation function, denoted as $KDF()$, and then encapsulated by the *encap()* algorithm using the client's public key, $cpk_t$, which has been stored in the server's database since the *enrolment phase*. the encapsulated key is delivered to the client and can be correctly decapsulated by running the *decap()* algorithm, only if the reconstruction of $csk_{t'} = csk_t$ is successful.

---

Verification

| Client | Server | Evaluator |
|---|---|---|
| $t'$—probe template | $ssk \in K$ | $k \in K$ |
| $id$—client's identifier | $spk \in P$ | |
| $spk \in P$ | $(V, cpk_t, id)$ | |

$$\xrightarrow{\quad id \quad}$$

$$\xleftarrow{\quad V \quad}$$

$f' \leftarrow unlock(V, t')$
$(B', r) \leftarrow blind(f')$
$(csk_e, cpk_e) \leftarrow KeyGen(1^\lambda)$ $\qquad\qquad (ssk_e, spk_e) \leftarrow KeyGen(1^\lambda)$

$$\xrightarrow{\quad B', cpk_e \quad} \qquad\qquad \xrightarrow{\quad B' \quad}$$

$\qquad\qquad\qquad (ctx, \gamma) \leftarrow encap(cpk_t) \qquad\qquad S' \leftarrow eval(B', k)$
$\qquad\qquad\qquad \rho \leftarrow KDF(cpk_t, spk, cpk_e, spk_e, \gamma)$

$$\xleftarrow{\quad S', spk_e, ctx, H(\rho) \quad} \qquad\qquad \xleftarrow{\quad S' \quad}$$

$csk_{t'} \leftarrow unblind(S', r)$
$cpk_{t'} \leftarrow pkGen(csk_{t'})$
$\gamma' \leftarrow decap(ctx, csk_{t'})$
$\rho' \leftarrow KDF(cpk_{t'}, spk, cpk_e, spk_e, \gamma')$
Return: $H(\rho') = H(\rho)$

---

**Figure 5.** Scheme of the *verification phase* introduced in BRAKE, based on [2].

*3.6. BRAKE Protocol Correctness*

The correctness of the protocol is based on the need for the client to obtain the secret key, $csk_{t'} = csk_t$, as a result of the *fuzzy vault* and OPRF primitives. Taking into account how both keys are generated, it is evident that for their equivalence, it is necessary to recover the coefficients of the secret polynomial, $f'(x) = f(x)$, from the polynomial, $V(x)$, sent to the client at the beginning of the *verification phase* by the server. Looking at the idea of using the *fuzzy vault* primitive, the conclusion is that obtaining $f'(x) = f(x)$ is possible only when the vault, $V(x)$, is unlocked correctly, which also requires the user to provide a biometric template, $t'$, which meets the condition $|t \cap t'| \geqslant \tau$. Unlocking the vault can be achieved by determining the set of pairs $\{(b, V(b)) : \forall b \in t'\}$, and then finding the most frequently yielded polynomial, $f'(x)$, as a result of the Lagrange interpolation on $\tau$-point subsets. This is possible due to the fact that if $b \in |t \cap t'|$, then $V(b) = f(b)$. The authors also suggest using the Guruswami–Sudan decoder [7], treating polynomials as Reed–Solomon codes [8], allowing the correct decoding of the polynomial $V(x)$ to the form of $f'(x) = f(x)$.

Considering the above description, it is possible to define the BRAKE protocol correctness as follows:

$$
\begin{aligned}
H(\rho') = H(\rho) &\iff \\
\rho' = KDF(\ldots, decap(encap(pkGen(csk_t)), csk_{t'})) & \\
= KDF(\ldots, \gamma') = KDF(\ldots, \gamma) = \rho &\iff csk_{t'} = csk_t \\
&\iff f'(x) = f(x) \iff |t' \cap t| \geq \tau
\end{aligned}
\tag{5}
$$

## 4. Security of BRAKE Protocol Analysis

Due to the fact that a detailed analysis of the security, as well as the possible limitations of the BAKE protocol, were presented in [2], in this work, we focus on indicating potential threats resulting from the proposed structure of the BRAKE protocol. We also point out that the discussion of the BRAKE protocol security was conducted in the original work, but the included security proofs are not sufficient in the context of the obtained conclusions.

### 4.1. Threat of Compromising the Evaluator's Secret Key

The developers of the BRAKE protocol propose an implementation of the evaluator's party, in which the same value of the evaluation key, $k \in K$, is used for each $eval()$ operation, for each individual client. This is motivated by the fact that the use of a larger number of keys, for instance, assigning each serviced customer a key that corresponds exclusively to them would require the introduction of a user identification mechanism for the evaluator. This would also contradict the assumption that the evaluator should obtain no knowledge about the client being serviced, as covered in Section 3.3. This includes information about the owner of the evaluated value. This is an accurate observation, but only if there is the possibility of rotating the key, $k$, used by the evaluator because it cannot be assumed that the key is completely resistant to compromise. According to Figure 4, and the use of the OPRF primitive suggested by the authors' hashed Diffie–Hellman variant [6], the client's public key yielded during the *enrolment phase* has the following form:

$$cpk_t = pkGen(csk_t) = pkGen([k]H(f)) \xrightarrow{HashDH\ OPRF} cpk_t = pkGen(H(f)^k) \qquad (6)$$

The public key representation in Equation (6) directly indicates that in order to successfully authenticate the client, the keys $csk_{t'}$ and $cpk_t = pkGen(csk_t)$ have to be derived from the same evaluation key, $k$, value. Assuming a scenario in which the key, $k$, is compromised, the server is required to invalidate all client identities whose public key, $cpk_t$, was generated using $k$. As a consequence, all clients whose identities have been revoked are forced to go through the re-enrolment process. This is due to the fact that for a compromised evaluator key, $k$, an adversary can launch an offline attack against the identity of a selected user, knowing only their identifier, $id$.

Algorithm 1 depicts our proposition of the offline attack scheme. We also provide a short semantic description as follows.

To successfully launch an offline attack against a specific user, the adversary, $\mathcal{A}$, must know the values of the compromised evaluation key, $k$, as well as the identifier, $id$, of the user to be attacked. The first phase of the attack corresponds with impersonating the user of the identifier, $id$, and requesting an authentication procedure to the server. After receiving the victim's vault, $V_{id}(x)$, an adversary randomly generates $r_{\mathcal{A}} \in \mathbb{Z}_q$ and computes its hash value, $H(r_{\mathcal{A}})$. It is worth noting that $r_{\mathcal{A}}$ can be arbitrarily chosen since the evaluator and server have no way of checking whether the user's OPRF input was computed based on the $f'(x)$ coefficients or that it had been forged. Also, if the OPRF protocol is suggested by the BRAKE protocol, the authors hashed Diffie–Hellman, there is no need to obfuscate the adversary's input into $[r]H(r_{\mathcal{A}})$, as the evaluator has no mechanism to define whether the input values were obfuscated or not. The adversary also generates the ephemeral key pair $(csk_e, cpk_e) \leftarrow KeyGen(1^\lambda)$ and proceeds to the OPRF execution, providing the server with $cpk_e$ and the evaluator with $H(r_{\mathcal{A}})$. After the OPRF finishes, $\mathcal{A}$ yields $ctx$, $H(\rho)$, and $spk_e$ values, and stores the tuple $(V_{id}(x), H(\rho), ctx, cpk_e, spk_e)$ for further attack execution.

---

**Algorithm 1** Offline attack against the user's $t_{id}$ template, with compromised $k$

---

**Require:** Victim's identifier, $id$, compromised evaluation key, $k$

 1: Send authentication request using $id$ to the server
 2: Receive $V_{id}(x)$ from the server
 3: Randomly generate $r_{\mathcal{A}} \in \mathbb{Z}_q$ and compute the hash function value $H(r_{\mathcal{A}})$
 4: Generate ephemeral key pair $(csk_e, cpk_e) \leftarrow KeyGen(1^{\lambda})$
 5: Execute OPRF evaluation using $H(r_{\mathcal{A}})$ and share $cpk_e$ with the server
 6: Receive $ctx$, $H(\rho)$ and $spk_e$ from the server
 7: Store: $V_{id}(x), H(\rho), ctx, cpk_e, spk_e$
 8: Randomly generate the set of guess templates $T_{\mathcal{A}} = \{t_{\mathcal{A}_1}, t_{\mathcal{A}_2}, \dots\}$
 9: **for each** $t_{\mathcal{A}_i} \in T_{\mathcal{A}}$ **do**
10:     Unlock $V_{id}(x)$ into: $f'_{\mathcal{A}_i}(x) \leftarrow unlock(V_{id}(x), t_{\mathcal{A}_i})$
11:     Compute: $csk_{t_{\mathcal{A}_i}} \leftarrow H(f'_{\mathcal{A}_i}(x))^k$
12:     Generate public key: $cpk_{t_{\mathcal{A}_i}} \leftarrow pubGen(csk_{t_{\mathcal{A}_i}})$
13:     Decapsulate pre-shared key: $\gamma_{\mathcal{A}_i} \leftarrow decap(ctx, csk_{t_{\mathcal{A}_i}})$
14:     Compute: $\rho_{\mathcal{A}_i} = KDF(cpk_{t_{\mathcal{A}_i}}, spk, cpk_e, spk_e, \gamma_{\mathcal{A}_i})$
15:     **if** $H(\rho_{\mathcal{A}_i}) = H(\rho)$ **then**
16:         **Return:** $t_{\mathcal{A}} = t_{\mathcal{A}_i}$
17:     **end if**
18: **end for**

---

The second phase of the attack focuses on finding the adversarial template, $t_{\mathcal{A}}$, that satisfies $|t_{\mathcal{A}} \cap t_{id}| \tau \geqslant$, which allows for the successful authentication and key exchange on the behalf of the victim, based on the BRAKE's correctness given in Equation (5). Firstly, the adversary generates a set $T_{\mathcal{A}} = \{t_{\mathcal{A}_1}, t_{\mathcal{A}_2}, \dots\}$ consisting of randomly generated adversarial guess templates, $t_{\mathcal{A}_i}$, where $|T_{\mathcal{A}}|$ strictly depends on the adversary's computing and storage capabilities. According to the statements presented in [2,5], for fingerprint-based *fuzzy vault* implementations, offline attacks pose a significant threat by lowering the number of operations required for an adversary to obtain a guess template that satisfies $|t_{\mathcal{A}} \cap t| \geqslant \tau$. This is also the reason why the BRAKE protocol authors decided to eliminate this threat by requiring user interaction with both server and evaluator instances during every authentication attempt. Since the evaluation key, $k$, has been compromised and is known to the adversary, it is possible to locally unlock the vault into $f'_{\mathcal{A}_i}(x) \leftarrow unlock(V_{id}(x), t_{\mathcal{A}_i})$, compute the secret key, $csk_{t_{\mathcal{A}_i}} \leftarrow H(f'_{\mathcal{A}_i}(x))^k$, and generate the public key, $cpk_{t_{\mathcal{A}_i}} \leftarrow pkGen(csk_{t_{\mathcal{A}_i}})$. Using the obtained secret key, $csk_{t_{\mathcal{A}_i}}$, the adversary attempts to decapsulate a pre-shared key as $\gamma_{\mathcal{A}_i} \leftarrow decap(ctx, csk_{t_{\mathcal{A}_i}})$. Since all the required values are known to the adversary, the guess session key is computed into $\rho_{\mathcal{A}_i} = KDF(cpk_{t_{\mathcal{A}_i}}, spk, cpk_e, spk_e, \gamma_{\mathcal{A}_i})$ and the attacker checks for $H(\rho_{\mathcal{A}_i}) = H(\rho)$. If the assumption of non-collision of the used hash function $H(\cdot)$ is held and the above equality is obtained, it means that the used guess template, $t_{\mathcal{A}_i}$, satisfies the condition $|t_{\mathcal{A}_i} \cap t| \geqslant \tau$, and the launched attack is successfully returning $t_{\mathcal{A}} = t_{\mathcal{A}_i}$. Using the template $t_{\mathcal{A}}$ yielded from Algorithm 1, the adversary can successfully impersonate the client of the identifier, $id$, until their identity is revoked from the server's database.

We want to highlight that even modifying the OPRF to use the *Updatable OPRF* [6] variation of the primitive for key rotation, $k$, has no effect and does not prevent the threat of offline attacks because it is not possible for the server to recover the value of $csk_t$ based on $cpk_t = pkGen(csk_t)$, which would be needed in order to make the key pair dependent on the rotated new evaluator's key value.

As a partial solution to the presented problem, we propose using functionality that allows the server and evaluator instances to split users into anonymized groups. In this case, the evaluator is not forced to use only one key, which when compromised, forces all registered users of the system to rerun the identity registration procedure. The proposed solution assumes that the evaluator uses different keys for each group, where the users have been divided. In this way, the evaluator can receive from the server an identifier of

the key to be used in the evaluation process, without gaining any additional information about the user for whom the evaluation is computed. This solution reduces the impact of compromising a single evaluation key by reducing the number of users whose identities must be invalidated. However, it is not an ideal solution, as it still prevents the continued secure use of identities created using a compromised key. In order to fully mitigate the threat presented in this section, it would be necessary to entirely change the scheme of generating asymmetric keys that are the basis for client authentication, but the proposal of such a solution is not within the scope of this work.

### 4.2. Threat of Secret Value Storage in Client's Device

Taking into account the user authentication scheme presented in [2] and depicted in Figure 5, it is apparent that the element certifying the client's identity is an asymmetric cryptography key pair created using *fuzzy vault* and OPRF primitives. According to the assumptions presented by the authors of the BRAKE protocol, all secret values (e.g., secret polynomials, asymmetric key pairs, hash values) should be stored in the volatile memory of the device, without the possibility of saving into persistent memory. However, the assumption of honest data processing by the client's device shown in BRAKE's threat model applies to the processing of biometric data only. In a scenario where the client has successfully completed the authenticated key exchange process but saves the private key, $csk_{t'}$, or the coefficients of the secret polynomial, $f'(x) = f(x)$, into the device's memory before establishing a communication session with the server, they can consistently pass the authentication process on each attempt, even if a sample template does not meet the $|t \cap t_{\mathcal{A}}| \geqslant \tau$ requirement. Assuming the widespread use of the BRAKE protocol in services that rely on a client–server communication model, we believe that the threat model presented in [2] should be expanded to include honest processing and not store any data type used by the client device, not limiting it to biometric data only. While biometrics are particularly sensitive due to the serious consequences of their potential compromise, according to the BRAKE protocol scheme, the possibility of storing any secret data used in it can lead to equivalent consequences. These consequences may result in the identity theft of the victim carried out by an adversary. However, service providers cannot control the devices of clients participating in the protocol to such an extent that they can ensure none of the secret values used to prove the user's identity have been stored in the device's persistent memory. For this reason, in order to mitigate this threat, it would be appropriate to consider modifying the BRAKE protocol with a mechanism that allows the server party to confirm the honesty of the client participating in its execution, including the honest participation in the OPRF protocol.

### 4.3. Threat of Client's Session Revocation

The *verification phase* of the BRAKE protocol, depicted in Figure 5, assumes the generation of a new symmetric key, $\rho$, whenever the server receives a request to verify a specific individual. This can be the source of a significant threat to the protocol. When a new session key, $\rho_i$, is generated, the previously used session key, $\rho_{i-1}$, is invalidated by the server instance. This can lead to a situation where an honestly authenticated client's session that uses the $\rho_{i-1}$ key is interrupted and revoked by an adversary forcing the generation of a new $\rho_i$ key, which claims to be an honest client. At no stage of the presented protocol is the state of the current session of a specific client, identified by *id*, checked. As authors of the protocol [2] have not addressed the above-presented threat, it is possible for an adversary to carry out an attack that consists of sending a verification request on behalf of a client with the identifier, *id*, which results in the revocation of access to the service by rotating the value of the session key from $\rho_{i-1}$ to $\rho_i$. It is worth noting that the only parameter that an adversary has to possess is the identifier, *id*, of the client to be attacked. By performing periodic queries to the server, the adversary is able to invalidate the client's session often enough to completely prevent the client with an honestly exchanged session key from conducting communication with the server.

Mitigating the above problem would require the use of mechanisms that allow for maintaining multiple sessions for a specific client at the same time. This would also apply to the use of multiple terminals simultaneously, such as communication with the server from both a mobile device and a workstation with an apparatus that allows for measuring and processing biometric characteristics. In addition, it would be important to implement a mechanism that allows invalidating the currently used session key only when the client has successfully completed the next authentication process.

*4.4. Threat of the Denial-of-Service Attack on the Server*

According to the *enrolment phase* scheme presented in Figure 4, it should be noted that at the end of its execution, the server does not validate the values received by the client in terms of confirming that the submitted data were honestly generated, moving through each step of the scheme. Thus, it is possible to design an attack scenario in which the adversary generates a falsified dataset, $(V_\mathcal{A}, cpk_\mathcal{A}, id_\mathcal{A})$, and requests the server to add it to the identity database. By executing a sufficient number of such requests for various values of identifiers, $id_\mathcal{A}$, it is possible to overflow the server's database. It should also be noted that the presented process of client enrolment allows the adversary to overload the server by sending a significant number of registration requests using falsified data, which can lead to significant consumption of the server's computational capability while entering the data into the mentioned database. This results in the occurrence of a *denial-of-service* attack, where honest clients are unable to carry out communication with the server due to its excessive overload.

A solution to this problem may be to introduce a limitation on the number of requests that can be forwarded by a specific device. Such a limitation has the effect of protecting the server instance from processing a significant number of requests received from a specific device during the *denial-of-service* attack execution process. An additional safeguard to mitigate the storage of unwanted, falsified client data in the server's database is the application of functionality for validating the $(V, cpk, id)$ dataset provided by the clients. However, its implementation method is not the subject of this work; hence, it will not be discussed in more detail within this content.

A *denial-of-service* attack can also be executed by an adversary during the *verification phase*, as part of the threat outlined in Section 4.3. Figure 5 shows that the server generates an asymmetric key pair $(ssk_e, spk_e)$, a new session key, $\rho$, and calculates the values of $ctx$ and $H(\rho)$ whenever a user authentication request is received. Assuming that all of the above operations are computationally demanding, with a sufficiently high number of falsified requests received from the adversary, it is possible to overload the computing power of the server instance, thereby preventing honest clients from communicating with the server.

## 5. Conclusions

This work presents the potential threats regarding the use of the *biometric resilient authenticated key exchange* protocol, as proposed in [2], as a result of the analysis conducted from the perspective of both the security of the protocol's idealized scheme and the risks arising from its improper implementation.

It has been shown that the protocol is not immune to the compromise of the secret key used by the evaluator party involved in the execution of the *oblivious pseudo-random function* primitive. This leads to the possibility of launching an offline attack against the client's biometric template, which is used as a user authentication factor in the BRAKE protocol. This threat is particularly significant because the whole idea behind the creation of the BRAKE protocol was based on the requirement for the aforementioned primitive to interact with the client during every authentication attempt, which should effectively prevent this attack from being carried out.

Attention was also drawn to the threat caused by the possibility of a rogue user storing the secret values in their device's persistent memory, which could be used in the future to

dishonestly but successfully pass the verification process by using correct authentication key values, despite the delivery of a non-mated biometric modality. An important conclusion that emerges from the analysis carried out in Section 4.2 is the very superficial use of biometric authentication methods in the BRAKE protocol. Indeed, biometric authentication is not required for the successful key exchange between the server and the client, due to the confirmation of the client's identity based on the possession of an asymmetric key pair corresponding to the keys created during the client registration phase. Thus, one should strive to create a variation of the BRAKE protocol based primarily on biometric user authentication, which should be the basis for carrying out an authenticated cryptographic key exchange, without the possibility of a rogue user circumventing the process, as outlined in Section 4.2.

Our analysis also focused on the proper way of implementing the BRAKE protocol, pointing out the need to add mechanisms that allow for proper management of user sessions, preventing their invalidation by an adversary. It is also important to protect the server instance from the overload caused by receiving a significant number of requests that use falsified enrolment data, which can lead to the overflowing of the server's identity database. Adversaries may also cause a reduction in the efficiency of serving honest users as a result of launching a *denial-of-service* attack, as outlined in Sections 4.3 and 4.4. This also leads us to the finding for the cybersecurity solution, which should be sustainable [9]. It is important to remember that when analyzing cryptographic protocols, special attention should be paid not only to the correctness of their assumptions and ideological schemes but also to the ways in which they can be implemented in real-life systems, to prevent potential adversaries from launching attacks, addressing not only the possible compromise of the cryptographic system but also ensuring the reliability of the devices used for their implementation.

*Future Works*

The use of biometric authentication methods in cryptographic solutions is a broad and (in many aspects) unexplored field; hence, we believe that further research into algorithms and protocols that combine these two fields is indeed very forward-looking, especially given the high reliability of biometric identity proofs of individuals. However, as outlined in the framework of this work, in order to mitigate the significant risks identified in Sections 4.1 and 4.2, it would be necessary to modify the BRAKE protocol from the ground up, in terms of how it generates keys based on which users are authenticated, as well as provide mechanisms to prove their integrity. The strength of protocols based on biometric methods should come directly from the use of proven and well-studied solutions for biometric user authentication in such a way that it becomes impossible to circumvent such an important aspect of the protocol, according to the analysis undertaken in Sections 4.1 and 4.2.

**Author Contributions:** Conceptualization, M.G. and W.W.; Methodology, M.G. and W.W.; Validation, M.G.; Formal analysis, M.G.; Investigation, M.G.; Resources, M.G.; Writing—original draft, M.G.; Writing—review & editing, W.W.; Supervision, W.W.; Project administration, W.W.; Funding acquisition, W.W. All authors have read and agreed to the published version of the manuscript.

**Funding:** This research received no external funding.

**Data Availability Statement:** Data are contained within the article.

**Conflicts of Interest:** The authors declare no conflicts of interest.

## Abbreviations

The following abbreviations are used in this manuscript:

| | |
|---|---|
| AFEM | asymmetric fuzzy encapsulation mechanism |
| AKE | authenticated key exchange |
| BAKE | biometrics-authenticated key exchange |
| BRAKE | biometric-resilient authenticated key exchange |
| KEM | key encapsulation mechanism |
| OPRF | oblivious pseudo-random function |
| PPT | probabilistic polynomial time |
| TTP | trusted third party |

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
