# Peer review of "Analysis of Biometric-Based Cryptographic Key Exchange Protocols—BAKE and BRAKE"

_cryptography, doi:10.3390/cryptography8020014_

Round 1
Reviewer 1 Report
Comments and Suggestions for Authors
Biometric authentication is of great interest these days. The paper explores the use of biometric authentication methods in cryptographic solutions, focusing on the BAKE and BRAKE protocols. [The authors should spell out abbreviations the first time they are used, not just in a list of abbreviations at the end.]These protocols aim to enable secure key exchange between users and services on the Internet using biometric features for authentication. The analysis highlights the security vulnerabilities and limitations of the BRAKE protocol, emphasizing the need to address these issues for successful implementation in modern systems. The paper goes along with current thinking about the importance of biometric authentication in confirming individual identity and discusses the challenges of integrating fuzzy biometrics into cryptographic protocols. However, a reference or two supporting the idea would be good.
On line 89 Figure ?? seems to be missing.
Some of the figures do not add much to the text, for example Figure 2 seems to show the obvious.
Comments on the Quality of English LanguageThis was very hard to read. One thing that interfered with readability was many small mistakes. For example, the sentence from line 17 to 20 is very awkward. Another was minor spelling mistakes such as "Resilitent" on line 508. Line 221 is very unclear.
Author Response
Dear Reviewer,
Thank you for your comments and suggestions, we have taken them into consideration while revising our paper, and below we refer to the specific ones, including our responses.
Remark 1- Abbreviations were not spelled out
Thank you for pointing out the lack of an appropriate extension of the abbreviations used for the names of the protocols on which the work was based. We included abbreviated extensions in the text at the first occurrence of a given term as suggested – what has been edited especially in the Abstract and the Introduction section of the paper.
Remark 2- On line 90 Figure ?? seems to be missing
In the course of editing the article on the basis of the comments provided, we did not notice any errors in its content resulting from incorrect markings or LaTeX syntax errors. Further revision of the article indicated that the document provided should not contain the errors pointed out in the note, in accordance with the content of the document in PDF format. However, we would like to thank you for your vigilance and for highlighting a potential problem that may have arisen in connection with the manner in which the manuscript was submitted.
Remark 3 - Some of the figures do not add much to the text, for example Figure 2
Figure 2 showing a generic scheme of running a protocol based on the OPRF primitive is not only intended to illustrate to the reader what is the idea of the primitive, but – as we include point out in the article – it presents a contrast with the implementation of the primitive in the BRAKE protocol. The BRAKE protocol – according to the scheme presented by its creators – evaluates the client's input data on the side of the evaluator instance. This is the approach that distinguishes this implementation from the general idea of executing the protocol, through the need to obfuscate user’s input before it is sent to the evaluator. Therefore, we believe that the inclusion of the indicated Figure is important in the work, allowing it to be compared with the BRAKE scheme and to highlight the importance of the process of obfuscating client’s data.
Remark 4 - Difficulties in understanding the text due to the Quality of the English Language
Thank you very much for your comments on the reception of our work in terms of its readability and ease of understanding. These are particularly important to us because we want the explanation of our analysis and the conclusions obtained to be as understandable as possible for the reader. In the course of the manuscript revision, we took a close look at the examples indicated: the complex sentences on lines 17 to 20 and 221 and the spelling mistake on line 508. Based on the conclusions drawn, we reconstructed sentences that could cause difficulties in the reception of the work throughout the entire article.
Remark 5 - Insufficiency of the content and references used to introduce the reader to the subject matter covered by the article
According to the suggestion to increase the length and broaden the substantive content of the introduction concerning our work, we have divided the introduction into a larger quantity of paragraphs, splitting the content of the introduction into appropriate sections describing the reason and how the research was carried out. In revising the manuscript based on the received editorial suggestions, in the article's introduction, we indicated a greater number of citations for works thematically related to our article. We know the importance of presenting an appropriate background, allowing the reader to easily find more sources dealing with combining biometric methods with cryptographic solutions closely related to our work, so thank you very much for your feedback.
Kind regards,
Authors
Reviewer 2 Report
Comments and Suggestions for Authors
This manuscript investigates integrating biometric authentication methods into cryptographic solutions through the BAKE and BRAKE protocols. The primary objective is to tackle the challenge of precise authentication of individuals within digital systems. The protocols aim to bolster security and verify user identity without storing keys in persistent memory by deriving secret authentication keys directly from biometric modalities. The analysis uncovers vulnerabilities in the BRAKE protocol, such as susceptibility to offline attacks on biometric templates and potential compromise of asymmetric cryptography keys. Additionally, security risks associated with the storage of secret values and unauthorized interruptions in communication sessions are identified. The paper underscores the importance of effectively addressing these vulnerabilities to implement the protocols in modern systems. However, major revisions are needed before it can be accepted for publication.
1. The manuscript lacks algorithmic comparisons, which are essential for establishing the efficiency and authenticity of the developed algorithm. For instance, recent references should be compared, which could significantly enhance practicality and security. Adding a section carefully analyzing the efficiency comparison among various methods is recommended.
2. Some spelling or grammar errors need further scrutiny, such as "Page 2, Line 89, Figure ??"
3. Figure 1 is not in the manuscript; it starts directly from Figure 2. Please verify and correct accordingly.
Author Response
Dear Reviewer,
Thank you for your comments and suggestions, we have taken them into consideration while revising our paper, and below we refer to the specific ones, including our responses.
Remark 1 - Lack of algorithmic comparisons and the computational efficiency of the developed algorithm
We would like to thank you for pointing out the expected way to present the developed algorithm. We strongly agree with the need to estimate computational complexity and present the results of execution and simulation of solutions based on attacks on IT services. However, we would like to draw attention to the fact that the analysis carried out in the scope of our manuscript operates exclusively at the theoretical level. It does not include a discussion of specific, real-world applicable solutions and the creation of our own implementations of the algorithm that allows to launch an attack on a specific service and software implementation of the protocol. Thus, the algorithm presented in our paper that presents the procedure of executing an offline attack against a user's biometric template, analyzes the vulnerabilities of the BRAKE protocol in the conceptual layer. It is an theoretical algorithm based on pseudo-code. It concerns the possibility of using techniques that allow for a significant reduction in the computational complexity of attacks on biometric templates that use the Fuzzy Vault primitive based on the modality of fingerprints – indicated by the authors of the paper themselves.
Thus, we believe that the creation of an appropriate simulation and the analysis of the computational complexity of the algorithm proposed by us is not within the scope of our work, due to the theoretical nature of the analysis undertaken, which is summarized by our position presented above.
Remark 2 - Spelling mistakes and Quality of English Language
Thank you very much for your comments on the reception of our work in terms of its readability and ease of understanding. These are particularly important to us because we want the explanation of our analysis and the conclusions obtained to be as understandable as possible for the reader. During the manuscript revision, we looked closely at the construction of Sections and specific sentences. The revision of the article based on the provided feedback let us immediately spot and correct spelling mistakes in the work's contents. Based on the conclusions drawn, we reconstructed sentences that could cause difficulties in the reception of the work, throughout the entire article.
Remark 3 - Missing Figures – manuscript starting directly from Figure 2; Figure ?? on line 89
While editing the article based on the comments provided, we did not notice any errors in its content resulting from incorrect markings or LaTeX syntax errors – such as missing references or entire Figures included in the manuscript. Further revision of the article indicated that the document provided should not contain the errors pointed out in the note, in accordance with the document's content in PDF format. However, we would like to thank you for your vigilance and for highlighting a potential problem concerning how the manuscript was submitted.
Remark 4 - Insufficiency of content and references to introduce the reader to the subject matter covered by the article
According to the suggestion to increase the length and broaden the substantive content of the introduction concerning our work, we have divided the introduction into a larger quantity of paragraphs, splitting the content of the introduction into appropriate sections describing the reason and how the research was carried out. In revising the manuscript based on the received editorial suggestions, in the article's introduction, we indicated a greater number of citations for works thematically related to our article. We know the importance of presenting an appropriate background, allowing the reader to easily find more sources dealing with combining biometric methods with cryptographic solutions closely related to our work, so thank you very much for your feedback.
Remark 5 - Unclear presentation of the work’s results
In order to increase the readability of the analysis and the conclusions obtained, we modified the form of presenting the generic algorithm for launching an offline attacks against the user's biometric templates. To achieve that the algorithm was presented both in a pseudocode algorithmic block available in LaTeX that allows for a clear specification of the sequence of steps taken in its implementation, as well as a semantic description of the idea behind the proposed algorithm. In accordance with the content of the previous paragraphs, as part of the revision of the article, changes and improvements were made in the quality of the English language used, which we also believe significantly increased the readability and ease of understanding of the results obtained as part of the analysis
Kind regards,
Authors
Round 2
Reviewer 1 Report
Comments and Suggestions for Authors
You have addressed all of my concerns and it is ready for publication.
Reviewer 2 Report
Comments and Suggestions for Authors
The authors have made satisfactory revisions to their manuscripts based on my previous criticisms. Therefore, I propose to publish this manuscript.